# Long Time Simulation Analysis of Geometry Dynamics Model under Iteration

**Weiwei Sun [1], Long Bai [1],\*****, Xinsheng Ge [1] and Lili Xia [2]**

[1] Mechanical Electrical Engineering School, Beijing Information Science and Technology University, Beijing 100192, China; sww@bistu.edu.cn (W.S.); gebim@vip.sina.com (X.G.)
[2] School of Applied Science, Beijing Information Science and Technology University, Beijing 100192, China; xll2004@126.com
\* Correspondence: bailong0316jn@126.com

**Abstract:** Geometry modeling methods can conserve the geometry characters of a system, which helps the dynamic equations more concisely and is good for long simulations. Reduced attitude, Lie group and Lie algebra are three different expressions of geometry. Models for the dynamics of a planer pendulum and a 3D pendulum were built with these three geometry expressions. According to the variation method, the dynamics models as ordinary differential equations were transformed into nonlinear equations which are solved by Newton iteration. The simulation results show that Lie group and Lie algebra calculations can conserve the geometric structure, but have different long-time behavior. The complete Lie group expression has the best long simulation behavior and has the lowest sensitivity to the time step in both planer and 3D pendulum simulations, because it saves the complete geometry of the system in the dynamics model.

**Keywords:** geometry mechanical; dynamics; lie group; Lie algebra; Newton iteration



## 1. Introduction

Computational geometric mechanics can keep the geometric characteristics of a system and have higher stability than traditional methods. In addition, the geometry-based model are more concise than trigonometric functions, which has incomparable advantages for spatial rigid–flexible coupling multi-body systems. Geometric methods are also used in the fields of informatics and information geometry [1], which a provides unified mathematical basis for the integration of mechanics and other disciplines. Therefore, the exploration of geometric representations, numerical calculations and physical significance will promote the engineering applications of geometric mechanics.

Geometric mechanics exploration has two directions: symplectic geometry and Lie group methods. The symplectic geometry method is used to solve differential algebraic equations of dynamic systems which combine dynamic analysis and differential equation solving. The discrete Lie group variational integrator is another geometry method which can conserve the energy, momentum and geometric structure. It is a recursive equation which can be used in the optimal control of a space dynamics system. In recent years, Shi et al. [2] studied the indirect symplectic method for solving optimal control problems of differential algebraic equations, which improved calculation efficiency, and Peng et al. [3] established the discrete Singer formula for optimal control of compressed bodies based on the Lagrange–D'Alembert principle and discrete variational principle, which reduced the number of drives required for optimal control. Peng, H.J. et al. [4] studied a symplectic optimal control method for robot trajectory control and obtained a symplectic discrete format for differential algebraic equations based on the discrete variational principle. Hu et al. [5] studied the relationship between symmetry rupture and energy dissipation in dynamics based on multi-symplectic theory. Hu et al. [6] explored structure conservation based on local structure conservation of the multi-symplectic method and established

a dynamic model of the flexible damping plate-spring mass system. These show the symplectic geometry method has developed towards the direction of multi-symplectic theory, mainly based on the conservation of local symplectic structure.

Differently from symplectic method, Lie group method seeks global conservation of geometry structure of system. Recent research achievements of Lie group are as follows: Leitz et al. [7] studied the polysymplectic Galerkin Lie group of geometrically accurate beams. Tariverdi et al. [8] explored flexible continuous manipulator modeling using a Lie group variational integrator. Sjoberg et al. [9] studied the dynamic model of a crane system based on Lie group theory. Mueller et al. [10] studied the application of spinor and Lie group theory in a recursive algorithm for a topologically tree-based multi-body system. Zdravko et al. [11] studied the forward dynamics of fixed-wing aircraft on the Lie group and realized the attitude non-singular reconstruction on SO(3). Zdravko et al. [12] studied the momentum and energy conservation Lie group integral method for rigid body rotational dynamics. Liu et al. [13] studied the finite-time optimal control of dynamic systems on Lie groups. Lee et al. [14–16] studied a Lie group variational integrator for full rigid body systems. Demoures et al. [17] studied a Lie group variational integrator for geometrically accurate beam dynamics.

In recent years, the moving frame method has provided a new idea for solving dynamics equations based on Lie group theory. The main results are: Zenkov et al. [18] studied Hamel form and variational integrators on a sphere. Shi et al. [19] studied the Hamel form under classical field theory and used the variational integrator in sledge model [20]. He also studied the minimum time optimal control based on a Hamel product molecule [21] and the Hamel form of infinite dimensional mechanical system [22]. Wang et al. [23] established the Hamel field variational product molecule of geometrically accurate beams. Wang et al. [24] carried out path planning for a manipulator based on a geometric mechanics method.

The above analysis shows that the discrete Lie group variational integrator and the Hamel method both change an ordinary differential equation into nonlinear equations for a numerical solution. Lie algebra works in the vector space which is mapped from a Lie group. The three methods solve the dynamics equations from different perspectives. The Hamel and Lie algebra methods reduce the equations' complexity. Based on the above explorations, some problems need to be analyzed. Firstly, the numerical simulations in the above explorations were performed with their own modeling methods, so the relation of different geometry expressions should be analyzed. Secondly, the extension of complexity of different geometry models and their numerical solutions should be analyzed. Thirdly, the numerical accuracy and stability of different geometry models need to be compared. The above analysis can help the engineer chose the best method to solve their own problem. Thus, in this paper, the geometric models for planar and 3D pendulums are established from three perspectives as reduced attitude, Lie group and Lie algebra, respectively. Based on the three models, the dynamics equations were transformed into nonlinear equations, and the Jacobi matrices of different equations were obtained. The numerical solutions were carried out by Newton iteration. Three methods were compared and analyzed based on the numerical results. This study provides a basis for the study of geometric dynamics of multi-body systems.

The structure of this research is as follows. In the Sections 2 and 3, the concepts of a Lie group, Lie algebra and Newton iteration are introduced based on basic mathematical expressions. In the Section 4, three kinds of geometric dynamics model of planer pendulum are built. In the Section 5, four types of geometry dynamics model of 3D pendulums are built. In the Section 6, the simulation results of different geometric dynamics models are presented. The conclusions are summarized in the Section 7. The concrete derivation of each model is presented step by step, which should help the readers to understand the methods easily.

## 2. Lie Group and Lie Algebra

The Lie group and Lie algebra are abstract mathematical concepts and difficult to understand. Thus, in this section, the concepts are explained based on concrete examples. Consider a rotation problem in plane. The coordinate transformation is expressed as in Figure 1. Suppose the initial coordinate is xOy, after a counterclockwise rotation with $\theta$. The new coordinate is x'Oy'. The rotation matrix between them is as shown in Equation (1).

$$R = \begin{bmatrix} \cos\theta & -\sin\theta \\ \sin\theta & \cos\theta \end{bmatrix} \tag{1}$$

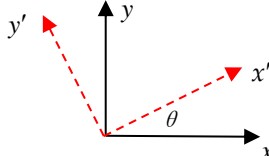

**Figure 1.** The coordinate transformation in plane.

The physical significance of the first and second column of $R$ is the projection of Ox' and Oy' on Ox and Oy respectively. $R$ can be written as the exponential type, as in Equation (2).

$$R = e^{S(\theta)} \tag{2}$$

In Equation (2), $S(\theta) = \begin{bmatrix} 0 & -\theta \\ \theta & 0 \end{bmatrix} = \theta \begin{bmatrix} 0 & -1 \\ 1 & 0 \end{bmatrix}$.

The derivation of $R$ is Equation (3).

$$\dot{R} = \begin{bmatrix} -\sin\theta & -\cos\theta \\ \cos\theta & -\sin\theta \end{bmatrix} \dot{\theta} = RS\left(\dot{\theta}\right) \tag{3}$$

The derivation of $R$ can be written as Equation (4) according to Equation (2).

$$\dot{R} = e^{S(\theta)} S\left(\dot{\theta}\right) \tag{4}$$

The results agree with the trigonometric functions type. Considering the system in space, the attitude matrices corresponding to the rotation along x, y, and z axes are in Equations (5)–(7), respectively.

$$R_x(\theta_3) = \begin{bmatrix} 1 & 0 & 0 \\ 0 & \cos\theta_1 & -\sin\theta_1 \\ 0 & \sin\theta_1 & \cos\theta_1 \end{bmatrix} \tag{5}$$

$$R_y(\theta_2) = \begin{bmatrix} \cos\theta_2 & 0 & \sin\theta_2 \\ 0 & 1 & 0 \\ -\sin\theta_2 & 0 & \cos\theta_2 \end{bmatrix} \tag{6}$$

$$R_z(\theta_1) = \begin{bmatrix} \cos\theta_3 & -\sin\theta_3 & 0 \\ \sin\theta_3 & \cos\theta_3 & 0 \\ 0 & 0 & 1 \end{bmatrix} \tag{7}$$

Derive Equations (5)–(7). The results are in Equation (8).

$$\dot{R}_x = \omega_x R_x S(e_1); \dot{R}_y = \omega_y R_y S(e_2); \dot{R}_z = \omega_z R_z S(e_3) \tag{8}$$

In Equation (8), $e_1 = [1; 0; 0], e_2 = [0; 1; 0], e_3 = [0; 0; 1]$.

$$S_{e_1} = \begin{bmatrix} 0 & 0 & 0 \\ 0 & 0 & -1 \\ 0 & 1 & 0 \end{bmatrix} S_{e_2} = \begin{bmatrix} 0 & 0 & 1 \\ 0 & 0 & 0 \\ -1 & 0 & 0 \end{bmatrix} S_{e_3} = \begin{bmatrix} 0 & -1 & 0 \\ 1 & 0 & 0 \\ 0 & 0 & 0 \end{bmatrix}$$

Equations (5)–(7) can also be written exponentially, as in Equation (9).

$$R_x = e^{\theta_1 S_{e_1}}; R_y = e^{\theta_2 S_{e_2}}; R_z = e^{\theta_3 S_{e_3}} \tag{9}$$

The derivation of Equation (9) is similar to Equation (4). $R$ can be seen as the Lie group expression of planer rotation through the concept of kinematics, and $\theta$ is the Lie algebra which corresponds to the Lie group. If $R$ represents the rotation in space, then the Lie algebra which corresponds to $R_x, R_y, R_z$ is $\theta e_1, \theta e_2, \theta e_3$.

According to the above analysis, $R$ is a function about $\theta$, no matter the triangle function or exponent type. The above theory can be used as the basis for modeling the dynamics of a single body or a series of single bodies combined together which rotate in space. In practical engineering, some rigid body has three degrees of rotational freedom, such as a ship, plane, spacecraft or UAV. They have triangle function expression based on the classical multibody theory. For example, with the Cardan expression, the rotation of a rigid body in space is as the series of X–Y–Z, as in Figure 2, and the rotation matrix can be written as (10).

$$R = R_x R_y R_z \tag{10}$$

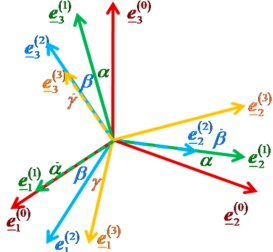

**Figure 2.** The Cardan rotation.

Differentiate $R$; then, substitute Equation (8) into Equation (10). The derivation of $R$ can be written as Equation (11).

$$\dot{R} = \dot{R}_x R_y R_z + R_x \dot{R}_y R_z + R_x R_y \dot{R}_z = \omega_x R_x S_x R_y R_z + \omega_y R_x R_y S_y R_z + \omega_y R_x R_y R_z S_z \tag{11}$$

Equation (11) can be rewritten as Equation (12) according to $R^T S(a) R = S(R^T a)$.

$$\begin{aligned} \dot{R} &= R_x R_y R_z (R_y R_z)^T S(\dot{\theta}_x e_1) R_y R_z + R_x R_y R_z R_z^T S(\dot{\theta}_y e_2) R_z + R_x R_y R_z S(\dot{\theta}_z e_3) \\ &= RS(R_z^T R_y^T \dot{\theta}_x e_1 + R_z^T \dot{\theta}_y e_2 + \dot{\theta}_z e_3) \end{aligned} \tag{12}$$

From the above equation, the rotation of a rigid body in space can be written as the nonlinear combination of the rotations on different axes Cardan order. Using the physics definition of $R$ in plane, the physics definition of each row of $R$ in space can be explained as the projection of X′Y′Z′ on X, Y, Z. Using $\omega$ to represent the absolute angular velocity of a rotation body in space, the differential of $R$ can be written as in Equation (13).

$$\dot{R} = RS(\omega) \tag{13}$$

In Equation (13), $\boldsymbol{\omega} = \begin{bmatrix} \omega_x & \omega_y & \omega_z \end{bmatrix}^T$. $\boldsymbol{S(\omega)}$ is the skew matrix of $\boldsymbol{\omega}$, as in Equation (14).

$$S(\boldsymbol{\omega}) = \begin{bmatrix} 0 & -\omega_z & \omega_y \\ \omega_z & 0 & -\omega_x \\ -\omega_y & \omega_x & 0 \end{bmatrix} \tag{14}$$

According to Equations (12) and (13), the angular velocity can be written as the combination of the rotation matrix and angular velocities across each axis, as in Equation (15).

$$\boldsymbol{\omega} = \boldsymbol{R}_z^T \boldsymbol{R}_y^T \dot{\theta}_x \boldsymbol{e}_1 + \boldsymbol{R}_z^T \dot{\theta}_y \boldsymbol{e}_2 + \dot{\theta}_z \boldsymbol{e}_3 \tag{15}$$

The above derivations are the expressions of the rotation matrix and their differentials of rigid body rotations in plane and in space. As part of dynamics solving, the above equations usually need to transform to the triangle type, which may make the expressions more complex due to the triangle transformation, especially for the rotation in space or multibody systems.

$$\|\boldsymbol{R}\| = 1; \boldsymbol{R}\boldsymbol{R}^T = \boldsymbol{I}; \boldsymbol{R}^T = \boldsymbol{R}^{-1} \tag{16}$$

In order to avoid these problems, the overall natures of Lie group and Lie algebra need to be analyzed. From the overall point of view, $\boldsymbol{R}$ satisfies the characteristics in Equation (16), no matter the plane or space.

Expect the above characteristics, each row of $\boldsymbol{R}$ also satisfies similar characteristics, as the moduli are one in each row.

$$\|\boldsymbol{R}_1\| = \|\boldsymbol{R}_2\| = \|\boldsymbol{R}_3\| = 1 \tag{17}$$

If each row of $\boldsymbol{R}$ is one element, Equation (13) can be divided into three parts, as in Equation (18).

$$\boldsymbol{e}_3^T \dot{\boldsymbol{R}} = \boldsymbol{e}_3^T \boldsymbol{R} S(\boldsymbol{\omega}); \boldsymbol{e}_2^T \dot{\boldsymbol{R}} = \boldsymbol{e}_2^T \boldsymbol{R} S(\boldsymbol{\omega}); \boldsymbol{e}_1^T \dot{\boldsymbol{R}} = \boldsymbol{e}_1^T \boldsymbol{R} S(\boldsymbol{\omega}) \tag{18}$$

Suppose $\boldsymbol{\Gamma}_i = \boldsymbol{R}^T \boldsymbol{e}_3$. Then, the derivation of $\boldsymbol{\Gamma}$ is

$$\dot{\boldsymbol{\Gamma}}_1 = S(\boldsymbol{\Gamma}_1)\boldsymbol{\omega}; \dot{\boldsymbol{\Gamma}}_2 = S(\boldsymbol{\Gamma}_2)\boldsymbol{\omega}; \dot{\boldsymbol{\Gamma}}_3 = S(\boldsymbol{\Gamma}_3)\boldsymbol{\omega} \tag{19}$$

For planer rotation, the solution of any part of Equation (19) can obtain the complete rotation matrix according to inverse derivation. For rotation in space, Equation (19) needs to be completely solved to obtain the rotation matrix. The above method avoids complex triangle transformations. The exponential type of rotation matrix is

$$\boldsymbol{R} = e^{S(\boldsymbol{\eta})} \tag{20}$$

$\boldsymbol{\eta}$ is the Lie algebra type. The exponential matrix can be obtained according to the Rodriguez parameter or Cayley transformation, as in Equations (21) and (22), respectively. The Cayley transformation avoids triangle expression completely.

$$\boldsymbol{R} = \boldsymbol{I} + \frac{\sin\|\boldsymbol{\eta}\|}{\|\boldsymbol{\eta}\|} S(\boldsymbol{\eta}) + \frac{1 - \cos\|\boldsymbol{\eta}\|}{\|\boldsymbol{\eta}\|^2} S^2(\boldsymbol{\eta}) \tag{21}$$

$$\boldsymbol{R} = \frac{\boldsymbol{I} + S(\boldsymbol{\eta})}{\boldsymbol{I} - S(\boldsymbol{\eta})} = \frac{1}{1 + \boldsymbol{\eta}^T\boldsymbol{\eta}} \left( \left(1 - \boldsymbol{\eta}^T\boldsymbol{\eta}\right)\boldsymbol{I} + 2S(\boldsymbol{\eta}) + 2\boldsymbol{\eta}\boldsymbol{\eta}^T \right) \tag{22}$$

The discussion of the exchangeability of the rotation matrix is as follows. Supposing that $\boldsymbol{R}_a = e^{\theta_a S(\boldsymbol{e}_i)}, \boldsymbol{R}_b = e^{\theta_b S(\boldsymbol{e}_j)}$, when $i = j$, $\boldsymbol{R}_a, \boldsymbol{R}_b$ is called isomorphism, which means it has the same structure. If and only if $i = j$, $\boldsymbol{R}_a, \boldsymbol{R}_b$ satisfies the relation as in Equation (23), which means that the rotation matrix has the same structure satisfying the exchange relation.

$$\boldsymbol{R}_a\boldsymbol{R}_b = \boldsymbol{R}_b\boldsymbol{R}_a = e^{\theta_a S(\boldsymbol{e}_i) + \theta_b S(\boldsymbol{e}_j)} \tag{23}$$

Suppose that the rotation matrix in space is $\boldsymbol{R} = e^{S(f)}$. The definitions of $\boldsymbol{R}_x, \boldsymbol{R}_y, \boldsymbol{R}_z$ are in Equation (9). According to Cardan relation, the exponential type of $\boldsymbol{R}$ is in Equation (24).

$$e^{S(f)} = e^{\theta_1 S_{e_1}} e^{\theta_2 S_{e_2}} e^{\theta_3 S_{e_3}} \tag{24}$$

It is necessary to declare $f \neq \begin{bmatrix} \theta_1 & \theta_2 & \theta_3 \end{bmatrix}^T$, which is important in the following derivation.

### 3. Newton Iterative

Traditionally, the dynamics equation which is ODE type is solved by the Runge–Kutta method. However, the error accumulation of this method may lead to the distortion of long simulations. The Lie group method and the Hamel method can avoid this problem. One reason is that modeling the dynamics of the geometry method can maintain the geometry characteristics of the system, and the other is that both of these methods solve the nonlinear equation, which is the coal character of these two methods. The distinguishing feature between the Lie group method and the Hamel method is that the Lie group method transforms the structure preserving part to be a nonlinear equation, and the Hamel method transforms the ODE to be a nonlinear equation directly. The nonlinear equation is often solved by Newtonian iteration. Thus, the error of each step can be reduced to be the lowest level and avoids error accumulation, which can guarantee the reliability of long simulations.

Suppose that the nonlinear equation is as Equation (25).

$$\begin{cases} f_1\begin{pmatrix} x_1, & x_2, & \ldots, & x_n \end{pmatrix} = 0 \\ f_2\begin{pmatrix} x_1, & x_2, & \ldots, & x_n \end{pmatrix} = 0 \\ \quad\quad \ldots\ldots\ldots\ldots \\ f_n\begin{pmatrix} x_1, & x_2, & \ldots, & x_n \end{pmatrix} = 0 \end{cases} \tag{25}$$

Using $\boldsymbol{x} = (x_1, x_2, \ldots, x_n)$, $\boldsymbol{F}(\boldsymbol{x}) = (f_1(\boldsymbol{x}), f_2(\boldsymbol{x}), \ldots, f_n(\boldsymbol{x}))^T$, Equation (25) can be written as a vector as in Equation (26).

$$\boldsymbol{F}(\boldsymbol{x}) = \boldsymbol{0} \tag{26}$$

Suppose that $\Delta\boldsymbol{x} = \boldsymbol{x} - \boldsymbol{x}^{(k)}$. $\boldsymbol{x}^{(k)}$ is the initial number of iterations. Substitute it into Equation (26). Then the linear equation system is Equation (28).

$$\boldsymbol{F}\left(\boldsymbol{x}^{(k)}\right) + \boldsymbol{F}'\left(\boldsymbol{x}^{(k)}\right)\Delta\boldsymbol{x}^{(k)} = 0 \tag{27}$$

Expanding Equation (28), we get the expression of $\boldsymbol{x}$.

$$\boldsymbol{x} = \boldsymbol{x}^{(k)} - \left[\boldsymbol{F}'\left(\boldsymbol{x}^{(k)}\right)\right]^{-1}\boldsymbol{F}\left(\boldsymbol{x}^{(k)}\right) \tag{28}$$

In Equation (29), $\boldsymbol{F}'\left(\boldsymbol{x}^{(k)}\right)$ is the Jacobi matrix of $\boldsymbol{F}\left(\boldsymbol{x}^{(k)}\right)$, which is written as Equation (30).

$$\boldsymbol{F}'\left(\boldsymbol{x}^{(k)}\right) = \begin{bmatrix} \frac{\partial f_1}{\partial x_1} & \frac{\partial f_1}{\partial x_2} & \cdots & \frac{\partial f_1}{\partial x_n} \\ \vdots & \vdots & & \vdots \\ \frac{\partial f_n}{\partial x_1} & \frac{\partial f_n}{\partial x_2} & \cdots & \frac{\partial f_n}{\partial x_n} \end{bmatrix}_{x=x^{(k)}} \tag{29}$$

According to continuous iteration under the given initial value, the exact solution of the system can be obtained which can satisfy the limiting condition. Then, the stop criterion of iteration is

$$\left\|\boldsymbol{x}^{(k+1)} - \boldsymbol{x}^{(k)}\right\| < \varepsilon \tag{30}$$

For example, Equation (18) can be discrete with Lie group or Lie algebra. Using the discrete model of a discrete Lie group variational integrator, the discrete rotation matrix can be written as Equation (32).

$$R_{k+\frac{1}{2}} = \frac{R_{k+1} + R_k}{2} \tag{31}$$

According to $\boldsymbol{\Gamma}_i = \boldsymbol{R}^T \boldsymbol{e}_i$, the discrete pattern on vector domain is shown in Equation (33).

$$\boldsymbol{\Gamma}_{ik+\frac{1}{2}} = \frac{\boldsymbol{\Gamma}_{ik+1} + \boldsymbol{\Gamma}_{ik}}{2} \tag{32}$$

$\boldsymbol{\Gamma}_{ik+1}$ can be obtained by the given $\boldsymbol{\Gamma}_{ik}$. The calculation can also use the difference as the unknown quantity, and the discrete schemes can be written as

$$\boldsymbol{\Gamma}_{ik+\frac{1}{2}} = \boldsymbol{\Gamma}_{ik} + \frac{\Delta \boldsymbol{\Gamma}_{ik}}{2} \tag{33}$$

Based on the given values of $\boldsymbol{\Gamma}_{ik}$ and $\Delta \boldsymbol{\Gamma}_{ik}$, the accuracy value of $\Delta \boldsymbol{\Gamma}_{ik}$ can be obtained according to the iterations, along with the accuracy value of $\boldsymbol{\Gamma}_{ik+1}$.

Suppose that there is a transfer matrix between $\boldsymbol{R}_{k+1}$ and $\boldsymbol{R}_k$. The discrete value of the rotation matrix is

$$\boldsymbol{R}_{k+1} = \boldsymbol{R}_k \boldsymbol{F}_k \tag{34}$$

Substitute Equation (35) into Equation (32) and get the expression of Equation (36).

$$R_{k+\frac{1}{2}} = \frac{R_{k+1} + R_k}{2} = \frac{1}{2} R_k (F_k + I) \tag{35}$$

The above expression can also transform to be exponential in type:

$$e^{S(\eta_{k+1})} = e^{S(\eta_k)} e^{S(\Delta \eta_k)} = e^{S(\eta_k + \Delta \eta_k)} \tag{36}$$

Equation (35) is the discrete form of a Lie group, so map the discrete relation to Lie algebra space. The discrete transformation can directly make Lie algebra space as it is in Equation (38). Thus, the goal of an iterative solution is $\Delta \boldsymbol{\eta}_k$ or $\boldsymbol{\eta}_{k+1}$ based on the known $\boldsymbol{\eta}_k$.

$$R_{k+\frac{1}{2}} = e^{S(\frac{\eta_k + \eta_{k+1}}{2})} = e^{S(\eta_k + \frac{\Delta \eta_k}{2})} \tag{37}$$

## 4. The Solution of the Dynamics of a Planer Pendulum

In this chapter, a planner pendulum is used to explain the above theory. The planner pendulum is shown in Figure 3, which considers its geometry. The mass and the rotational inertia of the pendulum are $m$ and $J$, respectively. The length between the center of mass and pivot is $l$. The dynamics model of the pendulum is written as Equation (39) with rotation matrix $\boldsymbol{R}$ and angular velocity $\boldsymbol{\omega}$.

$$\begin{cases} J\dot{\omega} + mgl e_2^T R e_1 = 0 \\ \dot{R} = RS(\omega) \end{cases} \tag{38}$$

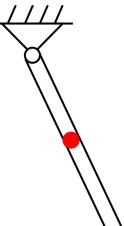

**Figure 3.** Planar pendulum.

Equation (39) can also be written as triangle type:

$$\begin{cases} J\dot{\omega} + mgl\sin\theta = 0 \\ \dot{\theta} = \omega \end{cases} \tag{39}$$

Let $\Pi = J\omega$. Then, Equation (39) can change to be of Hamilton type, as in Equation (41).

$$\begin{cases} \dot{\Pi} + mgl e_2^T \boldsymbol{R} e_1 = 0 \\ \dot{\boldsymbol{R}} = \boldsymbol{R}S\left(\frac{\Pi}{J}\right) \end{cases} \tag{40}$$

In general, Equation (41) can be solved by the Runge–Kutta method. Here, the difference method is used to solve it. The parameters are discrete:

$$\omega_{k+\frac{1}{2}} = \frac{\omega_{k+1} + \omega_k}{2}, \dot{\omega}_{k+\frac{1}{2}} = \frac{\omega_{k+1} - \omega_k}{h}, \dot{\theta}_{k+\frac{1}{2}} = \frac{\theta_{k+1} - \theta_k}{h}, \theta_{k+\frac{1}{2}} = \frac{\theta_{k+1} + \theta_k}{2} \tag{41}$$

Substitute Equation (42) into Equation (40). Then, the dynamics equation is

$$\begin{cases} J(\omega_{k+1} - \omega_k) + hmgl\sin\left(\frac{\theta_{k+1} + \theta_k}{2}\right) = 0 \\ 2(\theta_{k+1} - \theta_k) - h(\omega_{k+1} + \omega_k) = 0 \end{cases} \tag{42}$$

The Jacobi matrix of Equation (43) is

$$\begin{bmatrix} J & \frac{1}{2}hmgl\cos\left(\frac{\theta_{k+1} + \theta_k}{2}\right) \\ -h & 2 \end{bmatrix} \tag{43}$$

Discretize $\omega, \theta$ as Equation (45) to use $\Delta\omega_k$ and $\Delta\theta_k$ as parameters of a difference operation.

$$\dot{\omega}_{k+\frac{1}{2}} = \frac{\Delta\omega_k}{h}, \omega_{k+\frac{1}{2}} = \omega_k + \frac{\Delta\omega_k}{2}, \dot{\theta}_{k+\frac{1}{2}} = \frac{\Delta\theta_k}{h}, \theta_{k+\frac{1}{2}} = \theta_k + \frac{\Delta\theta_k}{2} \tag{44}$$

Substitute it into Equation (40) to obtain the dynamics equation as Equation (46), which uses $\Delta\omega_k, \Delta\theta_k$ as parameters.

$$\begin{aligned} J\Delta\omega_k + hmgl\sin\left(\theta_k + \frac{\Delta\theta_k}{2}\right) = 0 \\ 2\Delta\theta_k - 2h\omega_k - h\Delta\omega_k = 0 \end{aligned} \tag{45}$$

The Jacobi matrix of Equation (46) is

$$\begin{bmatrix} J & \frac{1}{2}hmgl\cos\left(\theta_k + \frac{\Delta\theta_k}{2}\right) \\ -h & 2 \end{bmatrix} \tag{46}$$

Let $\mathbf{\Gamma} = \mathbf{R}^T e_2$. Then, the Lie group expresses the dynamics equation as Equation (48).

$$\begin{cases} J\dot{\omega} + mgle_1^T\mathbf{\Gamma} = 0 \\ \dot{\mathbf{\Gamma}} = -S(\omega)\mathbf{\Gamma} \end{cases} \tag{47}$$

$\omega$ is discrete in a similar regular form as in Equation (42). The discrete regular of $\mathbf{\Gamma}$ is

$$\dot{\mathbf{\Gamma}}_{k+\frac{1}{2}} = \frac{\mathbf{\Gamma}_{k+1} - \mathbf{\Gamma}_k}{h}, \mathbf{\Gamma}_{k+\frac{1}{2}} = \frac{\mathbf{\Gamma}_{k+1} + \mathbf{\Gamma}_k}{2} \tag{48}$$

Substitute Equation (49) into Equation (48). The dynamics equation is

$$\begin{cases} 2J(\omega_{k+1} - \omega_k) + hmgle_1^T(\mathbf{\Gamma}_{k+1} + \mathbf{\Gamma}_k) = 0 \\ 4(\mathbf{\Gamma}_{k+1} - \mathbf{\Gamma}_k) + hS(\omega_{k+1} + \omega_k)(\mathbf{\Gamma}_{k+1} + \mathbf{\Gamma}_k) = 0 \end{cases} \tag{49}$$

Then, the Jacobi matrix is Equation (51), achieved by deriving $\omega_{k+1}, \mathbf{\Gamma}_{k+1}$.

$$\begin{bmatrix} 2J & hmgle_1^T \\ hS_1(\mathbf{\Gamma}_{k+1} + \mathbf{\Gamma}_k) & 4I + hS(\omega_{k+1} + \omega_k) \end{bmatrix} \tag{50}$$

Using $\omega_k, \mathbf{\Gamma}_k$ and $\Delta\omega_k, \Delta\mathbf{\Gamma}_k$ as parameters, the discretion of $\omega$ is Equation (44), and the discretion of $\mathbf{\Gamma}$ is Equation (52).

$$\dot{\mathbf{\Gamma}}_{k+\frac{1}{2}} = \frac{\Delta\mathbf{\Gamma}_k}{h}, \mathbf{\Gamma}_{k+\frac{1}{2}} = \mathbf{\Gamma}_k + \frac{\Delta\mathbf{\Gamma}_k}{2} \tag{51}$$

Substitute Equation (52) into Equation (48). The dynamics equation of system is Equation (53).

$$\begin{cases} 2J\Delta\omega_k + hmgle_1^T(2\mathbf{\Gamma}_k + \Delta\mathbf{\Gamma}_k) = 0 \\ 4\Delta\mathbf{\Gamma}_k + h(2\omega_k + \Delta\omega_k)S_1(2\mathbf{\Gamma}_k + \Delta\mathbf{\Gamma}_k) = 0 \end{cases} \tag{52}$$

The Jacobi matrix of Equation (53) is a $3 \times 3$ matrix:

$$\begin{bmatrix} 2J & hmgle_1^T \\ hS_1(2\mathbf{\Gamma}_k + \Delta\mathbf{\Gamma}_k) & 4I + h(2\omega_k + \Delta\omega_k)S_1 \end{bmatrix} \tag{53}$$

If $\mathbf{R}$ is discrete by Equations (35) and (36), and $\omega$ is discrete as in Equation (45), then substitute them into Equation (39) to get the dynamics equation of Equation (55).

$$\begin{cases} 2J\Delta\omega_k + hmgle_2^T R_k F_k e_1 + hmgle_2^T R_k e_1 = 0 \\ 2F_k e_1 - hF_k S\left(\omega_k + \frac{\Delta\omega_k}{2}\right)e_1 - 2e_1 - hS\left(\omega_k + \frac{\Delta\omega_k}{2}\right)e_1 = 0 \\ 2F_k e_2 - hF_k S\left(\omega_k + \frac{\Delta\omega_k}{2}\right)e_2 - 2e_2 - hS\left(\omega_k + \frac{\Delta\omega_k}{2}\right)e_2 = 0 \end{cases} \tag{54}$$

The latter two equations can change into the vector equation via multiplying them by $e_1$ and $e_2$. Let $F_k e_1 = \mathbf{\Gamma}_a, F_k e_2 = \mathbf{\Gamma}_b$. Equation (55) changes to be Equation (56).

$$\begin{cases} 2J\Delta\omega_k + hmgle_2^T R_k\mathbf{\Gamma}_a + hmgle_2^T R_k e_1 = 0 \\ 4\mathbf{\Gamma}_a - h(2\omega_k + \Delta\omega_k)\mathbf{\Gamma}_b - 4e_1 - h(2\omega_k + \Delta\omega_k)e_2 = 0 \\ 4\mathbf{\Gamma}_b + h(2\omega_k + \Delta\omega_k)\mathbf{\Gamma}_a - 4e_2 + h(2\omega_k + \Delta\omega_k)e_1 = 0 \end{cases} \tag{55}$$

$\Delta\omega_k, \mathbf{\Gamma}_a, \mathbf{\Gamma}_b$ are the variables of Equation (56). The Jacobi matrix is a $5 \times 5$ matrix, as shown in Equation (57).

$$\begin{bmatrix} 2J & hmgle_2^T R_k & \mathbf{0}^T \\ h(\mathbf{\Gamma}_b - e_2) & 4e_{12}^T & -h(2\omega_k + \Delta\omega_k)e_{12}^T \\ h(\mathbf{\Gamma}_a + e_1) & h(2\omega_k + \Delta\omega_k)e_{12}^T & 4e_{12}^T \end{bmatrix} \tag{56}$$

In Equation (57), $e_{12} = [1; 1]$. The above method changes $R$ into vector $\Gamma$. Although the numerical result is correct, the complete character of $R$ is broken, which may lead to constraint loss of the system. In order to reflect the complete structure of $R$, let $R^T e_1 = \Gamma_a$, $R^T e_2 = \Gamma_b$; then, Equation (39) can be written as Equation (58).

$$\begin{cases} J\dot{\omega} + mgle_1^T \Gamma_b = 0 \\ \dot{\Gamma}_a = -S(\omega)\Gamma_a \\ \dot{\Gamma}_b = -S(\omega)\Gamma_b \end{cases} \tag{57}$$

The discrete regulars of $\Gamma$ and $\omega$ are Equation (49) and Equation (42), respectively. With these regulars, the discrete dynamics equation is Equation (59).

$$\begin{cases} 2J(\omega_{k+1} - \omega_k) + hmgle_1^T(\Gamma_{b,k+1} + \Gamma_{b,k}) = 0 \\ 4(\Gamma_{a,k+1} - \Gamma_{a,k}) + h(\omega_{k+1} + \omega_k)S_1(\Gamma_{a,k+1} + \Gamma_{a,k}) = 0 \\ 4(\Gamma_{b,k+1} - \Gamma_{b,k}) + h(\omega_{k+1} + \omega_k)S_1(\Gamma_{b,k+1} + \Gamma_{b,k}) = 0 \end{cases} \tag{58}$$

It is necessary to notice that the significance of parameters in Equation (59) is different from those in Equation (56). If the equation is discrete with $\Delta\omega_k, \Delta\Gamma_{a,k}, \Delta\Gamma_{b,k}$, the result is Equation (60).

$$\begin{cases} 2J\Delta\omega_k + 2hmgle_1^T\Gamma_{b,k} + hmgle_1^T\Delta\Gamma_{b,k} = 0 \\ 4\Delta\Gamma_{a,k} + h(2\omega_k + \Delta\omega_k)S_1(2\Gamma_{a,k} + \Delta\Gamma_{a,k}) = 0 \\ 4\Delta\Gamma_{b,k} + h(2\omega_k + \Delta\omega_k)S_1(2\Gamma_{b,k} + \Delta\Gamma_{b,k}) = 0 \end{cases} \tag{59}$$

If the parameters are $\omega_{k+\frac{1}{2}}\Gamma_{a,k+\frac{1}{2}}\Gamma_{b,k+\frac{1}{2}}$, Equations (59) and (60) have the same Jacobi matrix as Equation (61).

$$\begin{bmatrix} 2J & 0_{1\times 2} & hmgle_1^T \\ 2hS_1\Gamma_{a,k+\frac{1}{2}} & 4I + 2h\omega_{k+\frac{1}{2}}S_1 & 0_{2\times 2} \\ 2hS_1\Gamma_{b,k+\frac{1}{2}} & 0_{2\times 2} & 4I + 2h\omega_{k+\frac{1}{2}}S_1 \end{bmatrix} \tag{60}$$

The above dynamics equations have Lie group expression. Next, the dynamics equation with Lie algebra is discussed. Firstly, the dynamics equation is changed to be the exponential type:

$$\begin{cases} J\dot{\omega} + mgle_2^T e^{S(\theta)}e_1 = 0 \\ \dot{\theta} = \omega \end{cases} \tag{61}$$

The discrete dynamics equation is shown in Equations (63) and (64). The discrete regulars are shown in Equations (42) and (45), respectively.

$$\begin{cases} J(\omega_{k+1} - \omega_k) + hmgle_2^T e^{S(\frac{\theta_{k+1}}{2})}e^{S(\frac{\theta_k}{2})}e_1 = 0 \\ 2(\theta_{k+1} - \theta_k) - h(\omega_{k+1} + \omega_k) = 0 \end{cases} \tag{62}$$

$$\begin{cases} J\Delta\omega_k + hmgle_2^T R_k e^{S(\frac{\Delta\theta_k}{2})}e_1 = 0 \\ 2\Delta\theta_k = 2h\omega_k + h\Delta\omega_k \end{cases} \tag{63}$$

The Jacobi matrixes corresponding to Equations (63) and (64) are in Equations (65) and (66).

$$\begin{bmatrix} J & \frac{hmgl}{2}e_2^T e^{S(\frac{\theta_{k+1}}{2})}e^{S(\frac{\theta_k}{2})}e_1 \\ -h & 2 \end{bmatrix} \tag{64}$$

$$\begin{bmatrix} J & \frac{hmgl}{2}e_2^T R_k e^{S(\frac{\Delta\theta_k}{2})}e_1 \\ -h & 2 \end{bmatrix} \tag{65}$$

The exponential part of Equations (63) and (64) can be factorized by the Cayley transformation as Equation (67).

$$R = \frac{1}{1+\theta^2}\left(\left(1-\theta^2\right)I + 2S(\theta) + 2\theta^2 e_1 e_1^T\right) \tag{66}$$

In the above analysis, the dynamics equations of a planer pendulum were built on different points. The dynamics equation with Lie algebra expression is more concise, and from the dimension point of view, the calculation based on Lie group theory needs more dimensions than is common. Although the dynamics equation has higher dimensionality, it can avoid the complex triangle transformation.

## 5. Dynamics Modeling of a 3D Pendulum

Based on the dynamics analysis of a planer pendulum, the dynamics modeling for a 3D pendulum was derived with Lie group and Lie algebra theory. The model of a 3D pendulum is shown in Figure 4. The 3D pendulum means that the pendulum has three degrees of freedom which can rotate along the pivot $O$ in space, and C is the center of mass. $\rho_C$ is the position vector of the center of mass. The dynamics equation of 3D pendulum is Equation (68).

$$J_C\dot{\omega}_C + S(\omega_C)\cdot J_C\omega_C - m_C g S(\rho_C)R_C^T e_3 = \mathbf{0}_{3\times 1}$$
$$\dot{R}_C - R_C S(\omega_C) = \mathbf{0}_{3\times 3} \tag{67}$$

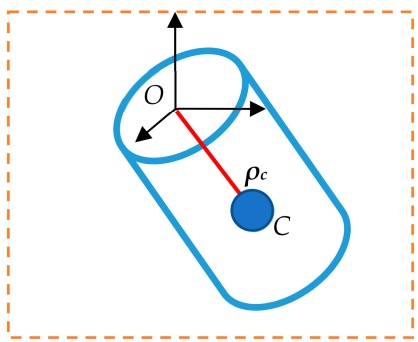

**Figure 4.** 3D rigid pendulum.

$J_C$ is the rotational inertia, $\omega_C$ is angular velocity along the inertia's principle axis and $R_C$ is the attitude matrix. Using $\Delta\omega_{C,k}, F_{C,k}$ as variables, the discrete regular is as Equation (69), which is similar to Equations (35) and (36).

$$\omega_{C,k+\frac{1}{2}} = \omega_{C,k} + \frac{\Delta\omega_{C,k}}{2}, R_{C,k+1} = R_{C,k}F_{C,k}, \dot{\omega}_{C,k+\frac{1}{2}} = \frac{\Delta\omega_{C,k}}{h},$$
$$R_{C,k+\frac{1}{2}} = \frac{1}{2}R_{C,k}(F_{C,k}+I), \dot{R}_{C,k+\frac{1}{2}} = \frac{1}{h}R_{C,k}(F_{C,k}-I) \tag{68}$$

Let $\dot{R}_C^T e_3 = -S(\omega_C)R_C^T e_3$. Substitute Equation (69) into Equation (68), and the dynamics equation of central difference type is Equation (70).

$$K_1\Delta\omega_{C,k} + hS(\Delta\omega_{C,k})J_C\Delta\omega_{C,k} - K_4 F_{C,k}^T K_3 + K_2 = \mathbf{0}$$
$$\left(K_6 F_{C,k}^T + hS(\Delta\omega_{C,k})\left(F_{C,k}^T + I\right)\right)K_3 + K_5 = 0 \tag{69}$$

$$K_1 = 4J_C - 2hS(J_C\omega_{C,k}) + 2hS(\omega_{C,k})J_C, K_2 = 4hS(\omega_{C,k})J_C\omega_{C,k} - 2hm_C g S(\rho_C)K_3,$$
$$K_3 = R_{C,k}{}^T e_3, K_4 = 2hm_C g S(\rho_C), K_5 = 2hS(\omega_{C,k})K_3 - 4K_3, K_6 = 4I + 2hS(\omega_{C,k})$$

Let $F_{C,k}^T K_3 = X$. Substitute it into Equation (70) to obtain the simplifier dynamics equation.

$$K_1\Delta\omega_{C,k} + hS(\Delta\omega_{C,k})J_C\Delta\omega_{C,k} - K_4 X + K_2 = \mathbf{0}$$
$$K_6 X + hS(\Delta\omega_{C,k})X + hS(\Delta\omega_{C,k})K_3 + K_5 = 0 \tag{70}$$

Then, the dynamics equation changes to be a nonlinear equation with variables of $\Delta\omega_{C,k}$ and $X$. In order to obtain the Jacobi matrix, the variation of Equation (71) is Equation (72).

$$
\begin{cases}
K_1\delta\Delta\omega_{C,k} + hS(\delta\Delta\omega_{C,k})J_C\Delta\omega_{C,k} + hS(\Delta\omega_{C,k})J_C\delta\Delta\omega_{C,k} - K_4\delta X = 0 \\
K_6\delta X + hS(\delta\Delta\omega_{C,k})X + hS(\Delta\omega_{C,k})\delta X + hS(\delta\Delta\omega_{C,k})K_3 + K_5 = 0
\end{cases}
\tag{71}
$$

Next, we change Equation (72) to matrix type:

$$
H\begin{bmatrix} \delta\Delta\omega_{C,k} \\ \delta X \end{bmatrix} = \mathbf{0}_{6\times 1}
\tag{72}
$$

$H$ is a Jacobi matrix of Equation (71), which is a $6 \times 6$ matrix.

$$
H = \begin{bmatrix}
K_1 + hS(\Delta\omega_{C,k})J_C - hS(J_C\Delta\omega_{C,k}) & -K_4 \\
-hS(X + K_3) & K_6 + hS(\Delta\omega_{C,k})
\end{bmatrix}
\tag{73}
$$

The reduced dynamics equation is Equation (75).

$$
\begin{aligned}
J_C\dot{\omega}_C + S(\omega_C)\cdot J_C\omega_C - m_C g S(\rho_C)\Gamma_C &= \mathbf{0} \\
\dot{\Gamma}_C &= -S(\omega_C)\Gamma_C
\end{aligned}
\tag{74}
$$

Define the discrete regular as Equation (76).

$$
\begin{aligned}
\Gamma_{C,k+\frac{1}{2}} &= \tfrac{1}{2}(\Gamma_{C,k+1} + \Gamma_{C,k}), \dot{\Gamma}_{C,k+\frac{1}{2}} = \tfrac{1}{h}(\Gamma_{C,k+1} - \Gamma_{C,k}), \\
\omega_{C,k+\frac{1}{2}} &= \tfrac{1}{2}(\omega_{C,k+1} + \omega_{C,k}), \dot{\omega}_{C,k+\frac{1}{2}} = \tfrac{1}{h}(\omega_{C,k+1} - \omega_{C,k})
\end{aligned}
\tag{75}
$$

Then, the dynamics equation with $\omega_{C,k+1}$ and $\Gamma_{C,k+1}$ as variables is Equation (77).

$$
\begin{aligned}
&4J_C\omega_{C,k+1} + hS(\omega_{C,k+1})J_C\omega_{C,k+1} + hS(\omega_{C,k})J_C\omega_{C,k+1} + hS(\omega_{C,k+1})J_C\omega_{C,k} \\
&-2hm_C g S(\rho_C)\Gamma_{C,k+1} + Q_A = 0 \\
&4\Gamma_{C,k+1} + hS(\omega_{C,k+1})\Gamma_{C,k+1} + hS(\omega_{C,k+1})\Gamma_{C,k} + hS(\omega_{C,k})\Gamma_{C,k+1} + Q_B = 0
\end{aligned}
\tag{76}
$$

By combining the same item together, the nonlinear equation changes to Equation (78).

$$
\begin{aligned}
Q_C\omega_{C,k+1} - Q_D\Gamma_{C,k+1} + hS(\omega_{C,k+1})J_C\omega_{C,k+1} + Q_A &= \mathbf{0} \\
Q_E\Gamma_{C,k+1} + hS(\omega_{C,k+1})(\Gamma_{C,k+1} + \Gamma_{C,k}) + Q_B &= \mathbf{0}
\end{aligned}
\tag{77}
$$

The expression levels of parameters $Q_A, Q_B, Q_C, Q_D, Q_E$ in Equation (78) are as follows.

$$
\begin{aligned}
&Q_A = hS(\omega_{C,k})J_C\omega_{C,k} - 4J_C\omega_{C,k} - 2hm_C g S(\rho_C)\Gamma_{C,k}, Q_B = hS(\omega_{C,k})\Gamma_{C,k} - 4\Gamma_{C,k}, \\
&Q_C = 4J_C + hS(\omega_{C,k})J_C - hS(J_C\omega_{C,k}), Q_D = 2hm_C g S(\rho_C), Q_E = 4I + hS(\omega_{C,k})
\end{aligned}
$$

Equation (79) is the variant of Equation (78) of matrix type.

$$
T\begin{bmatrix} \delta\omega_{C,k+1} \\ \delta\Gamma_{C,k+1} \end{bmatrix} = \mathbf{0}
\tag{78}
$$

$T$ is the Jacobi matrix in Equation (80).

$$
T = \begin{bmatrix}
Q_C - hS(J_C\omega_{C,k+1}) + hS(\omega_{C,k+1})J_C & -Q_D \\
-hS(\Gamma_{C,k+1} + \Gamma_{C,k}) & Q_E + hS(\omega_{C,k+1})
\end{bmatrix}
\tag{79}
$$

Similarly, if using $\Delta\mathbf{\Gamma}_{C,k}\Delta\boldsymbol{\omega}_{C,k}$ as parameters to discretize, the dynamics equation and Jacobi matrix are Equations (81) and (82), respectively.

$$
\begin{aligned}
\mathbf{W}_1\Delta\boldsymbol{\omega}_{C,k} + h\mathbf{S}(\Delta\boldsymbol{\omega}_{C,k})\mathbf{J}_C\Delta\boldsymbol{\omega}_{C,k} - \mathbf{W}_5\Delta\mathbf{\Gamma}_{C,k} + \mathbf{W}_2 = \mathbf{0} \\
-\mathbf{W}_6\Delta\boldsymbol{\omega}_{C,k} + \mathbf{W}_3\Delta\mathbf{\Gamma}_{C,k} + h\mathbf{S}(\Delta\boldsymbol{\omega}_{C,k})\Delta\mathbf{\Gamma}_{C,k} + \mathbf{W}_4 = \mathbf{0}
\end{aligned}
\tag{80}
$$

$$
\begin{bmatrix}
\mathbf{W}_1 - h\mathbf{S}(\mathbf{J}_C\Delta\boldsymbol{\omega}_{C,k}) + h\mathbf{S}(\Delta\boldsymbol{\omega}_{C,k})\mathbf{J}_C & -\mathbf{W}_5 \\
-\mathbf{W}_6 - h\mathbf{S}(\Delta\mathbf{\Gamma}_{C,k}) & h\mathbf{S}(\Delta\boldsymbol{\omega}_{C,k}) + \mathbf{W}_3
\end{bmatrix}
\tag{81}
$$

The parameters are as follows.

$$
\begin{aligned}
&\mathbf{W}_1 = 4\mathbf{J}_C - 2h\mathbf{S}(\mathbf{J}_C\boldsymbol{\omega}_{C,k}) + 2h\mathbf{S}(\boldsymbol{\omega}_{C,k})\mathbf{J}_C, \mathbf{W}_2 = 4h\mathbf{S}(\boldsymbol{\omega}_{C,k})\mathbf{J}_C\boldsymbol{\omega}_{C,k} - 4hm_Cg\mathbf{S}(\boldsymbol{\rho}_C)\mathbf{\Gamma}_{C,k}, \\
&\mathbf{W}_3 = 2h\mathbf{S}(\boldsymbol{\omega}_{C,k}) + 4\mathbf{I}, \mathbf{W}_4 = 4h\mathbf{S}(\boldsymbol{\omega}_{C,k})\mathbf{\Gamma}_{C,k}, \mathbf{W}_5 = 2hm_Cg\mathbf{S}(\boldsymbol{\rho}_C), \mathbf{W}_6 = 2h\mathbf{S}(\mathbf{\Gamma}_{C,k}),
\end{aligned}
$$

The reduced dynamics only consider part of the Lie group, which cannot represent the whole structure. Next, the calculation with a complete Lie group is derived. According to Equation (20), the dynamics equation with a complete Lie group is obtained. The discrete equation using $\Delta\boldsymbol{\omega}_{C,k}\Delta\mathbf{\Gamma}_{3,k}\Delta\mathbf{\Gamma}_{2,k}\Delta\mathbf{\Gamma}_{1,k}$ as parameters is Equation (83).

$$
\begin{aligned}
\mathbf{T}_1\Delta\boldsymbol{\omega}_{C,k} + h\mathbf{S}(\Delta\boldsymbol{\omega}_{C,k})\mathbf{J}_C\Delta\boldsymbol{\omega}_{C,k} - \mathbf{T}_3\Delta\mathbf{\Gamma}_{3,k} + \mathbf{T}_2 = \mathbf{0} \\
(\mathbf{T}_4 + h\mathbf{S}(\Delta\boldsymbol{\omega}_{C,k}))\Delta\mathbf{\Gamma}_{3,k} - \mathbf{T}_8\Delta\boldsymbol{\omega}_{C,k} + \mathbf{T}_5 = \mathbf{0} \\
(\mathbf{T}_4 + h\mathbf{S}(\Delta\boldsymbol{\omega}_{C,k}))\Delta\mathbf{\Gamma}_{2,k} - \mathbf{T}_9\Delta\boldsymbol{\omega}_{C,k} + \mathbf{T}_6 = \mathbf{0} \\
(\mathbf{T}_4 + h\mathbf{S}(\Delta\boldsymbol{\omega}_{C,k}))\Delta\mathbf{\Gamma}_{1,k} - \mathbf{T}_{10}\Delta\boldsymbol{\omega}_{C,k} + \mathbf{T}_7 = \mathbf{0}
\end{aligned}
\tag{82}
$$

The parameters are as follows.

$$
\begin{aligned}
&4\mathbf{J}_C + 2h\mathbf{S}(\boldsymbol{\omega}_{C,k})\mathbf{J}_C - 2h\mathbf{S}(\mathbf{J}_C\boldsymbol{\omega}_{C,k}) = \mathbf{T}_1, 4h\mathbf{S}(\boldsymbol{\omega}_{C,k})\mathbf{J}_C\boldsymbol{\omega}_{C,k} - 4hm_Cg\mathbf{S}(\boldsymbol{\rho}_C)\mathbf{\Gamma}_{3,k} = \mathbf{T}_2 \\
&2hm_Cg\mathbf{S}(\boldsymbol{\rho}_C) = \mathbf{T}_3, 4 + 2h\mathbf{S}(\boldsymbol{\omega}_{C,k}) = \mathbf{T}_4; 4h\mathbf{S}(\boldsymbol{\omega}_{C,k})\mathbf{\Gamma}_{3,k} = \mathbf{T}_5, 4h\mathbf{S}(\boldsymbol{\omega}_{C,k})\mathbf{\Gamma}_{2,k} = \mathbf{T}_6; \\
&4h\mathbf{S}(\boldsymbol{\omega}_{C,k})\mathbf{\Gamma}_{1,k} = \mathbf{T}_7, 2h\mathbf{S}(\mathbf{\Gamma}_{3,k}) = \mathbf{T}_8; 2h\mathbf{S}(\mathbf{\Gamma}_{2,k}) = \mathbf{T}_9, 2h\mathbf{S}(\mathbf{\Gamma}_{1,k}) = \mathbf{T}_{10}
\end{aligned}
$$

The Jacobi matrix is in Equation (84).

$$
\begin{bmatrix}
\mathbf{T}_1 + h\mathbf{S}(\Delta\boldsymbol{\omega}_{C,k})\mathbf{J}_C - h\mathbf{S}(\mathbf{J}_C\Delta\boldsymbol{\omega}_{C,k}) & \mathbf{0}_{3\times3} & \mathbf{0}_{3\times3} & -\mathbf{T}_3 \\
-h\mathbf{S}(\Delta\mathbf{\Gamma}_{3,k}) - \mathbf{T}_8 & \mathbf{0}_{3\times3} & \mathbf{0}_{3\times3} & \mathbf{Z} \\
-h\mathbf{S}(\Delta\mathbf{\Gamma}_{2,k}) - \mathbf{T}_9 & \mathbf{0}_{3\times3} & \mathbf{Z} & \mathbf{0}_{3\times3} \\
-h\mathbf{S}(\Delta\mathbf{\Gamma}_{1,k}) - \mathbf{T}_{10} & \mathbf{Z} & \mathbf{0}_{3\times3} & \mathbf{0}_{3\times3}
\end{bmatrix}
\tag{83}
$$

In Equation (84), $\mathbf{Z} = \mathbf{T}_4 + h\mathbf{S}(\Delta\boldsymbol{\omega}_{C,k})$. The dynamics analysis of Lie algebra is derived as follows. For the Lie group which corresponds to the 3D pendulum is not exchangeable, discretizing the Lie group directly is not available. Here, the Cardan transformation is used as the basis for the discrete use of Lie algebra. Combined with Equation (12), the dynamics equation is Equation (85).

$$
\begin{aligned}
\mathbf{J}_C\dot{\boldsymbol{\omega}}_C + \mathbf{S}(\boldsymbol{\omega}_C)\cdot\mathbf{J}_C\boldsymbol{\omega}_C - m_Cg\mathbf{S}(\boldsymbol{\rho}_C)(\mathbf{R}_x\mathbf{R}_y\mathbf{R}_z)^T\mathbf{e}_3 = \mathbf{0} \\
\boldsymbol{\omega}_C - \dot{\theta}_z(\mathbf{R}_y\mathbf{R}_x)^T\mathbf{e}_3 - \dot{\theta}_y\mathbf{R}_x^T\mathbf{e}_2 - \dot{\theta}_x\mathbf{e}_1 = \mathbf{0} \\
\boldsymbol{\omega}_C - \dot{\theta}_z\mathbf{e}_3 - \dot{\theta}_y\mathbf{R}_z^T\mathbf{e}_2 - \dot{\theta}_x(\mathbf{R}_y\mathbf{R}_z)^T\mathbf{e}_1 = \mathbf{0}
\end{aligned}
\tag{84}
$$

The discrete regular is defined in Equation (86).

$$
\begin{aligned}
&\boldsymbol{\omega}_{C,k+\frac{1}{2}} = \boldsymbol{\omega}_{C,k} + \frac{\Delta\boldsymbol{\omega}_{C,k}}{2}, \dot{\boldsymbol{\omega}}_{C,k+\frac{1}{2}} = \frac{\Delta\boldsymbol{\omega}_{C,k}}{h}; \dot{\theta}_{z,k+\frac{1}{2}} = \frac{\Delta\theta_{z,k}}{h}, \dot{\theta}_{y,k+\frac{1}{2}} = \frac{\Delta\theta_{y,k}}{h}, \dot{\theta}_{x,k+\frac{1}{2}} = \frac{\Delta\theta_{x,k}}{h} \\
&\mathbf{R}_{z,k+\frac{1}{2}} = e^{(\theta_{z,k} + \frac{\Delta\theta_{z,k}}{2})\mathbf{S}(\mathbf{e}_3)} = \mathbf{R}_{z,k}e^{\frac{\Delta\theta_{z,k}}{2}\mathbf{S}(\mathbf{e}_3)}; \mathbf{R}_{y,k+\frac{1}{2}} = e^{(\theta_{y,k} + \frac{\Delta\theta_{y,k}}{2})\mathbf{S}(\mathbf{e}_3)} = \mathbf{R}_{y,k}e^{\frac{\Delta\theta_{y,k}}{2}\mathbf{S}(\mathbf{e}_2)}; \\
&\mathbf{R}_{x,k+\frac{1}{2}} = e^{(\theta_{x,k} + \frac{\Delta\theta_{x,k}}{2})\mathbf{S}(\mathbf{e}_1)} = \mathbf{R}_{x,k}e^{\frac{\Delta\theta_{x,k}}{2}\mathbf{S}(\mathbf{e}_1)}
\end{aligned}
\tag{85}
$$

Substitute the former into Equation (86). The discrete dynamics equation of the system is then Equation (87).

$$
\begin{aligned}
&K\Delta\boldsymbol{\omega}_{C,k} + hS(\Delta\boldsymbol{\omega}_{C,k})\boldsymbol{J}_C\Delta\boldsymbol{\omega}_{C,k} + 4hS(\boldsymbol{\omega}_{C,k})\boldsymbol{J}_C\boldsymbol{\omega}_{C,k} \\
&-4hm_C g S(\boldsymbol{\rho}_C)\left(e^{-\frac{\Delta\theta_{x,k}}{2}S(e_1)}\boldsymbol{R}_{x,k}^T e^{-\frac{\Delta\theta_{y,k}}{2}S(e_2)}\boldsymbol{R}_{y,k}^T e^{-\frac{\Delta\theta_{z,k}}{2}S(e_3)}\boldsymbol{R}_{z,k}^T\right)e_3 = \boldsymbol{0} \\
&2h\boldsymbol{\omega}_{C,k} + h\Delta\boldsymbol{\omega}_{C,k} - 2\Delta\theta_{x,k}e_1 - 2\Delta\theta_{y,k}e^{-\frac{\Delta\theta_{x,k}}{2}S(e_1)}\boldsymbol{R}_{x,k}^T e_2 - 2\Delta\theta_{z,k}e^{-\frac{\Delta\theta_{x,k}}{2}S(e_1)}\boldsymbol{R}_{x,k}^T e^{-\frac{\Delta\theta_{y,k}}{2}S(e_2)}\boldsymbol{R}_{y,k}^T e_3 = \boldsymbol{0}
\end{aligned}
\tag{86}
$$

In Equation (87), $\boldsymbol{K} = 4\boldsymbol{J}_C + 2hS(\boldsymbol{\omega}_{C,k})\boldsymbol{J}_C - 2hS(\boldsymbol{J}_C\boldsymbol{\omega}_{C,k})$. The dimensions of Equation (87) number 6, and $\Delta\boldsymbol{\omega}_{C,k}\Delta\theta_{x,k}\Delta\theta_{y,k}\Delta\theta_{z,k}$ are their variables. The Jacobi matrix is Equation (87).

$$
\begin{bmatrix}
\boldsymbol{U}_{11} & \boldsymbol{U}_{12} & \boldsymbol{U}_{13} & \boldsymbol{U}_{14} \\
h\boldsymbol{I} & \boldsymbol{U}_{22} & \boldsymbol{U}_{23} & \boldsymbol{U}_{24}
\end{bmatrix}
\tag{87}
$$

$$
\boldsymbol{U}_{11} = \boldsymbol{K} + hS(\Delta\boldsymbol{\omega}_{C,k})\boldsymbol{J}_C - hS(\boldsymbol{J}_C\Delta\boldsymbol{\omega}_{C,k})
$$

$$
\boldsymbol{U}_{12} = 2hm_C g S(\boldsymbol{\rho}_C)e^{-\frac{\Delta\theta_{x,k}}{2}S(e_1)}S(e_1)\boldsymbol{R}_{x,k}^T e^{-\frac{\Delta\theta_{y,k}}{2}S(e_2)}\boldsymbol{R}_{y,k}^T e^{-\frac{\Delta\theta_{2k}}{2}S(e_3)}\boldsymbol{R}_{z,k}^T e_3
$$

$$
\boldsymbol{U}_{13} = 2hm_C g S(\boldsymbol{\rho}_C)e^{-\frac{\Delta\theta_{x,k}}{2}S(e_1)}\boldsymbol{R}_{x,k}^T e^{-\frac{\Delta\theta_{yk}}{2}S(e_2)}S(e_2)\boldsymbol{R}_{y,k}^T e^{-\frac{\Delta\theta_{2,k}k(e_3)}{2}}\boldsymbol{R}_{z,k}^T e_3
$$

$$
\boldsymbol{U}_{14} = 2hm_C g S(\boldsymbol{\rho}_C)e^{-\frac{\Delta\theta_{x,k}}{2}S(e_1)}\boldsymbol{R}_{x,k}^T e^{-\frac{\Delta\theta_{jyk}}{2}S(e_2)}\boldsymbol{R}_{y,k}^T e^{-\frac{\Delta\theta_{2,k}}{2}S(e_3)}S(e_3)\boldsymbol{R}_{z,k}^T e_3
$$

$$
\boldsymbol{U}_{22} = e^{-\frac{\Delta\theta_{x,k}}{2}S(e_1)}S(e_1)\boldsymbol{R}_{x,k}^T\left(\Delta\theta_{y,k}e_2 + \Delta\theta_{z,k}e^{-\frac{\Delta\theta_{y,k}}{2}S(e_2)}\boldsymbol{R}_{y,k}^T e_3\right) - 2e_1
$$

$$
\boldsymbol{U}_{23} = e^{-\frac{\Delta\theta_{x,k}}{2}S(e_1)}\boldsymbol{R}_{x,k}^T\left(\Delta\theta_{z,k}e^{-\frac{\Delta\theta_{y,k}}{2}S(e_2)}S(e_2)\boldsymbol{R}_{y,k}^T e_3 - 2e_2\right)
$$

$$
\boldsymbol{U}_{24} = -2e^{-\frac{\Delta\theta_{x,k}}{2}S(e_1)}\boldsymbol{R}_{x,k}^T e^{-\frac{\Delta\theta_{y,k}}{2}S(e_2)}\boldsymbol{R}_{y,k}^T e_3
$$

## 6. Simulation

Suppose that the mass of the pendulum is 2 kg, the rotational inertia is 0.48 kg·m², the length of the planer pendulum is 0.6 m, the initial angle is 0 rad, the initial angular velocity is 1 rad/s and the simulation time is 50 s. According to Equations (43) and (46), the simulation results are in Figures 5–8.

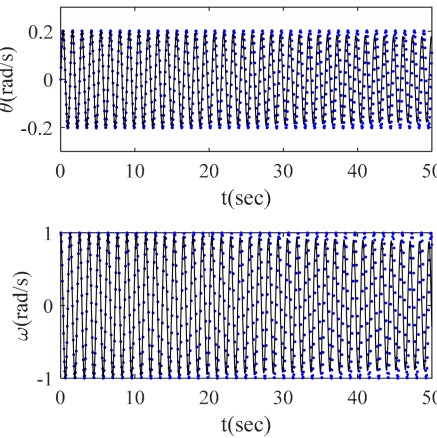

**Figure 5.** Comparison of 4th order Runge–Kutta and difference methods.

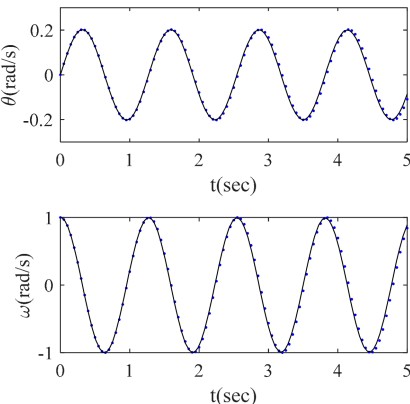

**Figure 6.** The comparison of the first 5 s.

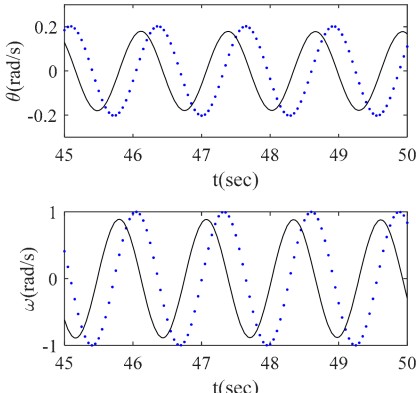

**Figure 7.** The comparison of the last 5 s.

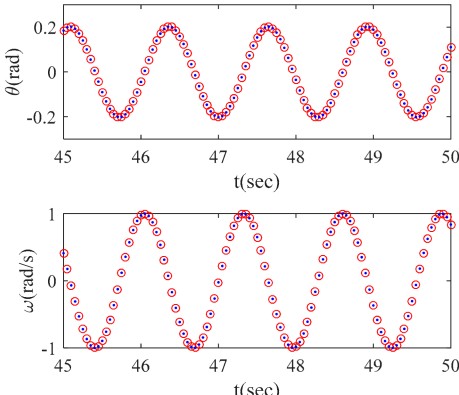

**Figure 8.** Comparison of two difference calculations.

According to Figure 5, the dynamics parameters of the planer pendulum are damped as the simulation time increases under the Runge–Kutta method. The simulation results under the difference method conserve the long-term stability of the attitude and angular velocity. The comparisons between the first and last 5 s for these two methods are shown in Figures 6 and 7, respectively. At the beginning of the calculation, the two results coincide, but in the last 5 s as in Figure 7, the differences between these two methods are obvious—not only the amplitude, but also the time period. Figure 8 shows a comparison of simulations under two difference methods, Equations (43) and (46), for the last 5 s. The simulation results highly coincide.

Figure 9 shows the simulation results of two difference calculation under reduced attitude. The simulation results also highly coincide. Figures 10 and 11 are the comparison

results of the difference calculations with reduced attitude or triangle expression with standard results, respectively. According to Figure 10, the three results highly coincide in the first 5 s, but the results appear obviously distinguishable from the standard values.

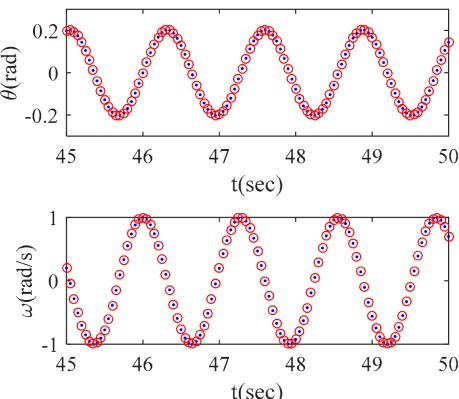

**Figure 9.** The difference calculation results under reduced attitude.

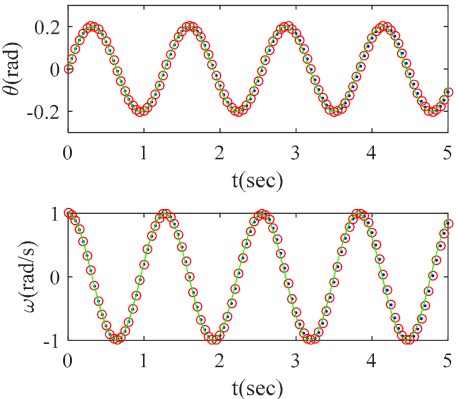

**Figure 10.** The results under reduced attitude and triangle function (1).

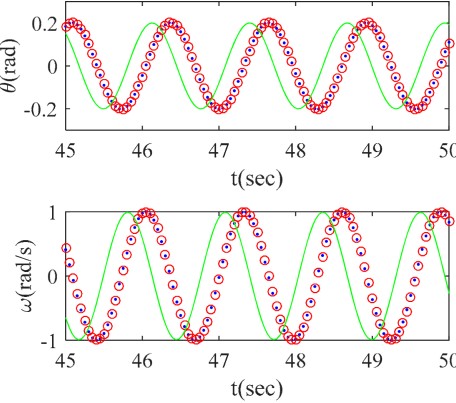

**Figure 11.** The results under reduced attitude and triangle function (2).

Figure 12 is a comparison of simulation results with different time steps and reduced attitude expression. When the time step is 0.05 s, the results are obviously distinguishable, but the results coincide when the time step is 0.01 s. This figure indicates that a long time step will lead to time hysteresis in the simulation.

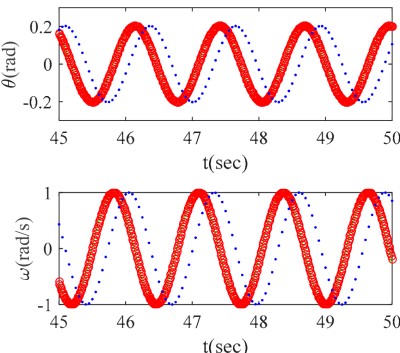

**Figure 12.** Simulation results with different time steps.

Figure 13 is a comparison between the simulation results according to Equation (56) and the standard results. The figure indicates that the results of the recursive method and the standard results highly coincide with a time step of 0.01 s.

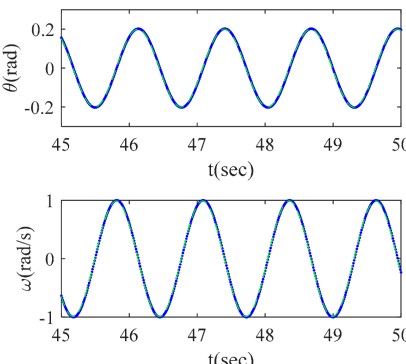

**Figure 13.** A comparison of the recursive method with the standard method.

Figures 14 and 15 are the simulation results which consider the complete Lie group structure. According to Figure 14, the two difference simulation results approximately coincide when the same big time step is used, 0.05 s, but the simulation results show some differentiation in the last 5 s. According to Figure 15, the simulation results which consider the complete Lie group structure highly coincide with the standard results when the time step is 0.01 s. Figures 16 and 17 are the comparisons of difference calculation results using Lie algebra. These two figures indicate that the two difference calculation results coincide when using same time step and also coincide with the standard value using the time step of 0.01 s.

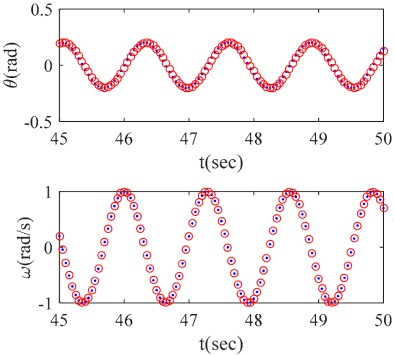

**Figure 14.** A comparison of two difference methods under the Lie group model with the time step of 0.05 s.

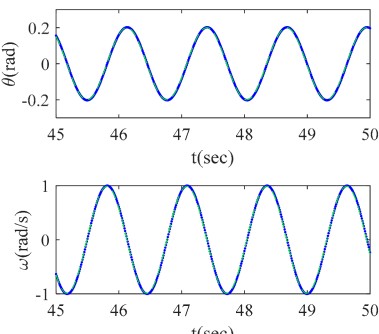

**Figure 15.** A comparison of two difference methods under the Lie group model with the time step of 0.01 s.

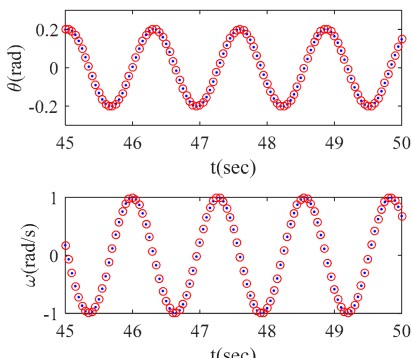

**Figure 16.** Two difference results of Lie algebra.

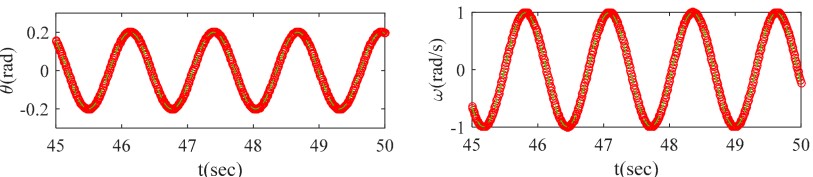

**Figure 17.** A comparison of the results using Lie algebra and standard results.

The dynamics calculations of a 3D pendulum are analyzed as follows. Supposing that the mass is 10 kg, the rotation inertias of the principle axes are $J_x$ = 1 kg·m$^2$; $J_y$ = 1 kg·m$^2$, $J_z$ = 1 kg·m$^2$. The position vector of the center of mass is [0;0;0.7] m, and the initial value of rotation matrix is $R_0$, which can derived by the Cardan method.

Supposing Cardan angles which correspond to attitude are $\theta_x = \frac{\pi}{3}, \theta_y = \frac{\pi}{4}, \theta_z = \frac{\pi}{9}$: the correspond rotation matrixes are $R_x, R_y, R_z$, respectively. Thus, the rotation matrix can be derived by $R_0 = R_x R_y R_z$. The initial angular velocity is $\omega = [0; 0; 0] \text{rad}/\text{s}$.

The dynamics calculation results based on Equation (70) are shown in Figures 18 and 19. The simulation time was 100 s. The time step was $h = 0.05$s. The simulation results of angular velocity and attitudes from 95to 100 s show good long-term stability under a large time step. Figures 20 and 21 are the simulation results of Equation (78), which is expressed by the reduced attitudes. The simulation time step was 0.1 s, and the simulation time was 100 s. According to the simulation results, the dynamics equation which is expressed by reduced attitudes has good long-term stability under Newton iteration with the time step of 0.1 s. This means that the reduced attitude expression has low sensitivity to the time step in the dynamics calculation. Figures 22–25 are the simulation results based on equation (83), which is expressed by the complete Lie group method. The simulation results indicate that the results of complete the Lie group method also have good long simulation stability under the time step of 0.1 s. Compared with

the reduced attitude expression, the complete Lie group method can obtain the results of all attitudes.

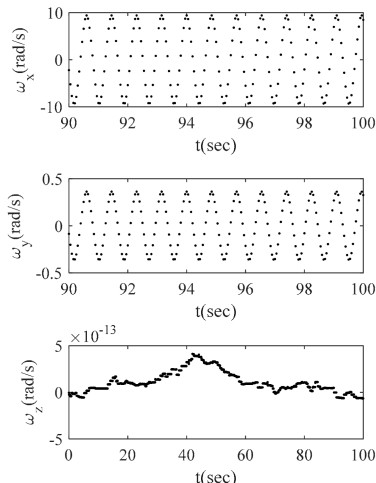

**Figure 18.** The simulation results of angular velocity.

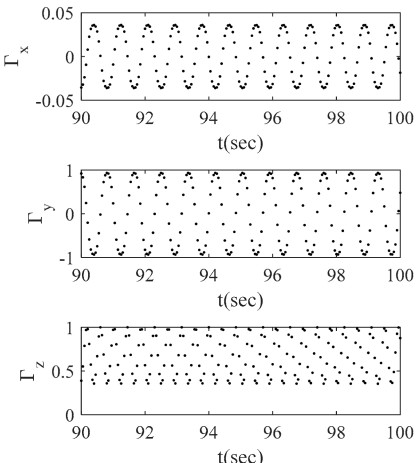

**Figure 19.** The simulation results of attitudes.

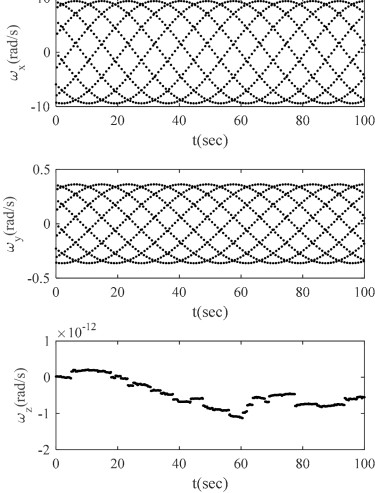

**Figure 20.** The angular velocity results under reduced attitude.

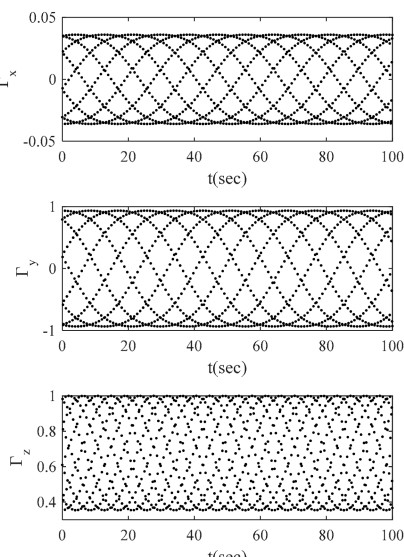

**Figure 21.** The attitude results under reduced attitude expression.

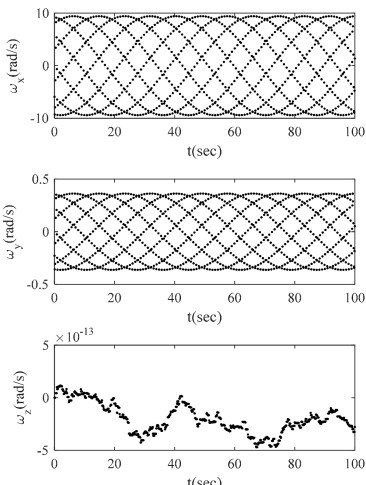

**Figure 22.** The angular velocity results under the complete Lie group model.

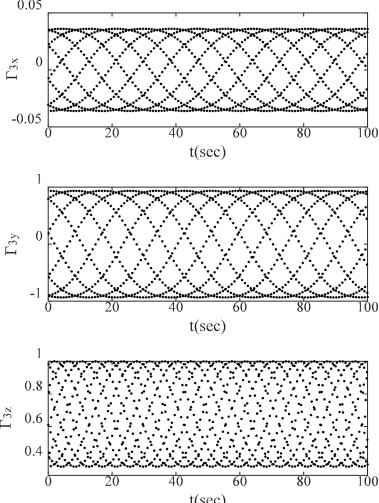

**Figure 23.** The results of the attitude of the *x* axis under the complete Lie group model.

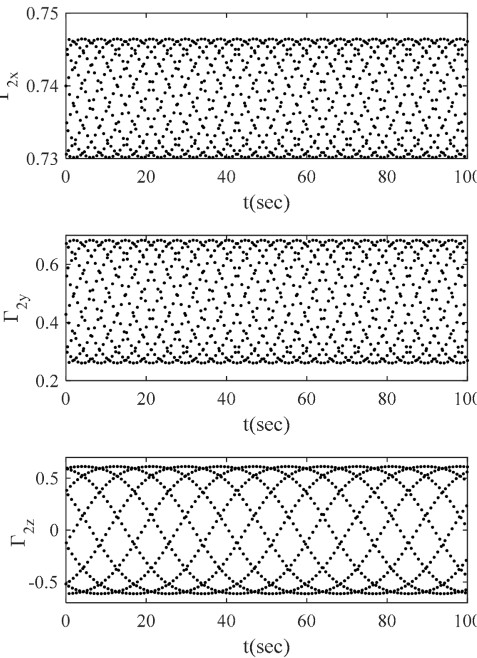

**Figure 24.** The results of the attitude of the *y* axis under the complete Lie group model.

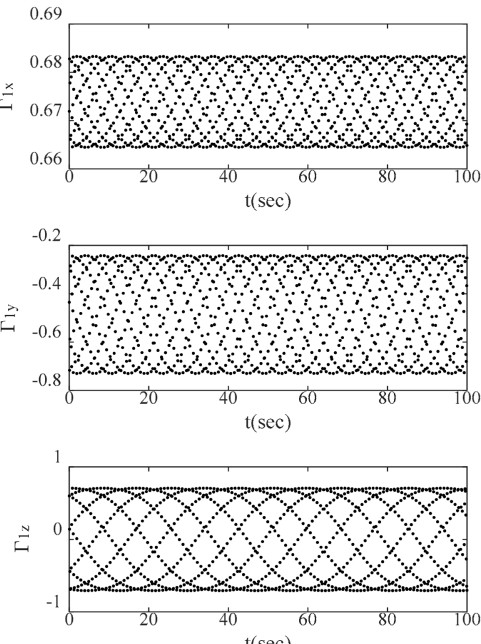

**Figure 25.** The results of the attitude of *z* axis under the complete Lie group model.

Figures 26 and 27 are simulation results of using Equation (87), which has the Lie algebra type expression. The simulation results were obtained with the time steps of 0.01 and 0.005 s. According to the simulation results, the dynamics results were lost under the Lie algebra, which is determined by Cardan rotation. It is very sensitivity to the time step.

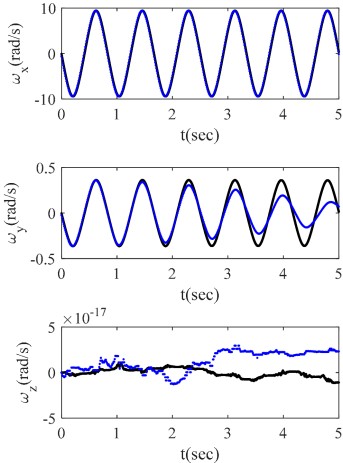

**Figure 26.** The angular velocity results under Lie algebra expression.

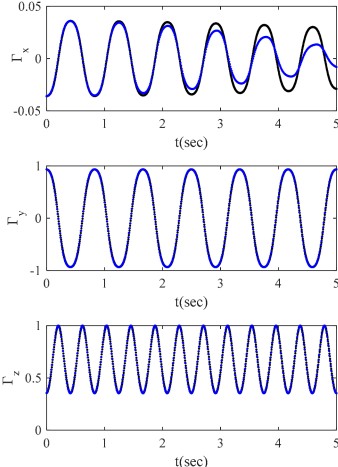

**Figure 27.** The attitude results under Lie algebra expression.

## 7. Conclusions

From the simulation results of a planer pendulum and a 3D pendulum, the conclusions can be summarized as follows.

(1) In the long simulation, the influence of error accumulation on the dynamics was not only reflected in the amplitude, but also in the periodic law of the system.

(2) The reduced attitude model of planer pendulum can conserve the long simulation characteristic with a special time step. If the time step is not suitable, the error accumulation may lead to the hysteresis of simulation. Thus, the error accumulation speed is obviously influenced by the time step.

(3) The complete Lie group dynamics model has similar simulation characteristics with different time steps, which means that the error accumulation is not sensitive to the time step. The simulation results using Lie algebra do not have this advantage.

(4) In the simulation of a 3D pendulum, the reduced attitudes and complete Lie group also had low sensitivity to the time step in the dynamics calculations. The simulation results for the time steps of 0.1 and 0.01 s are not obviously distinguishable.

(5) The dynamics results for the 3D pendulum lost their conservation under the Lie algebra model, which is determined by Cardan rotation, which means that the custom expression of rotation in space does not have the desired long simulation characteristics.

According to the above conclusions, the long simulation characteristic is influenced by the expression of the model and the time step simultaneously. The complete Lie group expression has the best long simulation characteristics and has the least sensitivity to the

time step for both planer and 3D pendulum simulations, because it conserves the complete geometry characteristics of the system.

**Author Contributions:** Conceptualization, X.G. and L.B.; methodology, L.B.; software, W.S.; validation, W.S.; formal analysis, L.B.; investigation, L.X.; resources, L.X.; data curation, W.S.; writing—original draft preparation, W.S.; writing—review and editing, L.B.; visualization, L.B.; supervision, L.X.; project administration, L.X.; funding acquisition, L.B. All authors have read and agreed to the published version of the manuscript.

**Funding:** This research was funded by Youth Fund of National Natural Science Foundation of China, grant number 11802035; National Natural Science Foundation of China, grant number 12072041; Major project of National Natural Science Foundation of China, grant number 11732005; General project of Science and Technology Plan of Beijing Municipal Education Commission, grant number KM201911232022; Talent support program of BISTU, grant number 5112111110.

**Institutional Review Board Statement:** Not applicable.

**Informed Consent Statement:** Not applicable.

**Data Availability Statement:** Not applicable.

**Conflicts of Interest:** The funders had no role in the design of the study; in the collection, analyses or interpretation of data; in the writing of the manuscript; or in the decision to publish the results.

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
