# Peer review of "Long Time Simulation Analysis of Geometry Dynamics Model under Iteration"

_applsci, doi:10.3390/app12104910_

Round 1

Reviewer 1 Report

Dear Authors,

there is room to improve the presentation of your work.

Please emphasize, based on the list of achievements in the introduction, what is the gap your paper fills in

Conclusions should be better supported by explanations (i.e. you say "It is very sensitivity to the time step." Please be more specific and try to explain how is the simulation affected by the increase and respectively decrease of the time step). Other similar discussions are welcome. 

Good luck!

Author Response

there is room to improve the presentation of your work.

Please emphasize, based on the list of achievements in the introduction, what is the gap your paper fills in

Answer the gap of achievements in the introduction is added in the updated paper to make the contribution of the exploration more clearly.

Firstly, the numerical simulations in the above former explorations are built on their own modeling method, so the relation of different geometry expressions should be analyzed. Secondly, the extension of complexity of different geometry models and their numerical solution should be analyzed. Thirdly, the numerical accuracy and stability of different geometry models need to be compared. The above analysis can help the engineer choose the best method to solve their own problem.

Conclusions should be better supported by explanations (i.e. you say "It is very sensitivity to the time step." Please be more specific and try to explain how is the simulation affected by the increase and respectively decrease of the time step). Other similar discussions are welcome. 

Good luck!

Answer: the conclusions are updated in the new version as follows.

From the simulation results of planer pendulum and 3D pendulum, the conclusions can be summarized as follows.

In the long time simulation, the influence of error accumulation to the dynamics not only reflected on the amplitude, but also on the periodic law of the system.

The reduced attitude model of planer pendulum can conserve the long time simulation character under special time step. If the time step is not suit, the error accumulation may lead to the hysteresis of simulation. So the error accumulation speed is obviously influenced by the time step.

The complete Lie group dynamics model has similar simulation characters under different time step, which means that the error accumulation is not sensitive to the time step. The simulation results under Lie algebra don’t have this advantage.

In the simulation of 3D pendulum, the reduced attitudes and complete Lie group also has a low sensitivity to the time step in the dynamics calculation. the simulation results under the time step of 0.1s and 0.01s have not obvious distinguish.

 The dynamics result of 3D pendulum is lost its conservation under the Lie algebra which is determinate by Cardan rotation, which means that the custom expression of rotation in space don’t have the long time simulation characters.

According to the above conclusions, the long time simulation character is influenced by the expression of model and the time step simultaneously. The complete Lie group expression has the best long time simulation character and has the lowest sensitive to the time step both in the planer and 3D pendulum simulation because it conserves the complete geometry characters of the system.

Reviewer 2 Report

Article “Long Time Simulation Analysis of Geometry Dynamics Model under Iteration” is in subject of Applied Sciences of MDPI journal.

Article present different geometry dynamics model and  results of its simulation.

Article need to solve some editorial problems presented in comments of reviewer.  Paper need to prepare more relevant conlusions about presented idea and results of its application

Comment 0

For improve an article readability I propose to add at the end of Introduction the paper structure

Comment 1

You wrote Section 2 and start from Figure 1

Need to add explanation what you plan in this section and need to cite of Figure 1 which you present before you it show. I dont find any explanation of Figure 1

Also Figure 2 is presented in Section 2 and not cited and explained

Comment 2

You wrote

  1. Newton Iterative

Traditionally, the dynamics equation which has the ODE type is solved by the Runge- Kutta method.

Can you add reference [?]

Comment 3

You wrote

  1. The dynamics solution of planer pendulum

Please change number of this section previous was numbered as 3.

Comment 4

In line

Figure 3 Planar pendulum

Not cited and explained in paper. Also you present this figure without explanation anything about this section.

Please improve it

Comment 5

Also next section was numbered as 3?

  1. Dynamics modeling of 3D pendulum

Please renumber it on 5

  1. Dynamics modeling of 3D pendulum

Please also describe of Figure 4 before it presentation in article

Comment 6

You wrote section 6 as

  1. Simulation

Please change number on 6

  1. Simulation

In this section you wrote

Supposing that the mass of the pendulum is 2kg, the rotation inertia is 0.48kgm2, the length of the planer pendulum is 0.6m, the initial angle is 0 rad, the initial angular velocity is 1rad/s, the simulation time is 50s. according to equation (43) and (46), the simulation results are as fig 5 to fig 8.

I propose use in paper  Fig. 5 to Fig. 8 and also please improve all “fig ” on “Fig. ”

Comment 7

Fig 7 The comparison of the later 5 seconds 400

Fig 8 Comparison of two difference calculation

Figure 7 The comparison of the later 5 seconds 400

Figure 8 Comparison of two difference calculation

And next

Fig 9 The difference calculation results under reduced attitude

Fig 10 The results under reduced attitude and triangle function(1)

Please change on

Figure 9 The difference calculation results under reduced attitude

Figure 10 The results under reduced attitude and triangle function(1)

Comment  8

I propose to use description of Figures 5 to 27 in column to clearly presents of relevant description to presented figures

Please also correct of bold letters used for

Fig 26 The angular velocity results under Lie algebra expression

Fig 27 The attitude results under Lie algebra expression

Comment  9

You wrote conclusions

  1. Conclusions

According to the above exploration, the long time simulation character is influenced by the modeling method, the numerical method synthetically. Different modeling and numerical method has different sensitive to time step size. The simulation shows that under the nonlinear equation solving method, the dynamics model which conserves the Lie group structure has a good long time simulation character and low sensitive to the time step.

In my opinion is possible to prepare more relevant conlusions to presented article.

Author Response

Comments and Suggestions for Authors

Article “Long Time Simulation Analysis of Geometry Dynamics Model under Iteration” is in subject of Applied Sciences of MDPI journal.

Article present different geometry dynamics model and  results of its simulation.

Article need to solve some editorial problems presented in comments of reviewer.  Paper need to prepare more relevant conlusions about presented idea and results of its application

Answer: The conclusions of this paper are updated as follows which can make the contribution of this paper more clearly.

From the simulation results of planer pendulum and 3D pendulum, the conclusions can be summarized as follows.

In the long time simulation, the influence of error accumulation to the dynamics not only reflected on the amplitude, but also on the periodic law of the system.

The reduced attitude model of planer pendulum can conserve the long time simulation character under special time step. If the time step is not suit, the error accumulation may lead to the hysteresis of simulation. So the error accumulation speed is obviously influenced by the time step.

The complete Lie group dynamics model has similar simulation characters under different time step, which means that the error accumulation is not sensitive to the time step. The simulation results under Lie algebra don’t have this advantage.

In the simulation of 3D pendulum, the reduced attitudes and complete Lie group also has a low sensitivity to the time step in the dynamics calculation. the simulation results under the time step of 0.1s and 0.01s have not obvious distinguish.

 The dynamics result of 3D pendulum is lost its conservation under the Lie algebra which is determinate by Cardan rotation, which means that the custom expression of rotation in space don’t have the long time simulation characters.

According to the above conclusions, the long time simulation character is influenced by the expression of model and the time step simultaneously. The complete Lie group expression has the best long time simulation character and has the lowest sensitive to the time step both in the planer and 3D pendulum simulation because it conserves the complete geometry characters of the system.

Comment 0

For improve an article readability I propose to add at the end of Introduction the paper structure(在简介部分增加对文章结构的描述)

Answer: The introduction of the structure of the paper is as follows which has already added in the revised version.

The structure of this research is as follows. In the second and third part, the concept of Lie group, Lie algebra and Newton iterative are introduced base on basic mathematical expression. In the forth part, three kinds of geometry dynamic model of planer pendulum are built. In the fifth part, four types geometry dynamics model of 3D pendulum are built. In the sixth part, the simulation results of different geometry dynamic models are presented. The conclusion is summarized in the last part. The concrete derivation of each model is presented step by step which can help the readers to understand the method easily. 

Comment 1

You wrote Section 2 and start from Figure 1

Need to add explanation what you plan in this section and need to cite of Figure 1 which you present before you it show. I dont find any explanation of Figure 1

Answer: the plan of this part and the explanation of Fig1 is added in the updated paper as follows.

The Lie group and Lie algebra are abstract mathematical concepts and difficult to be understand. So in this part , the concepts are explained base on the concrete examples. Consider a rotation problem in plane. The coordinate transformation is expressed as in Fig 1.

Also Figure 2 is presented in Section 2 and not cited and explained

Answer: the explanation of fig 2 is added in the updated version as follows.

For example, with Cardan expression, the rotation of rigid body in space is as the series of X-Y-Z, as in Fig 2.

Comment 2

You wrote

  1. Newton Iterative

Traditionally, the dynamics equation which has the ODE type is solved by the Runge- Kutta method.

Can you add reference [?]

Answer: in common use, the ODE often solved by Runge-Kutta method. if we discuss the essential problem of Runge-Kutta method, its the improved type of Euler method. The basic of Euler method is the discretization of the differential part of ODE equation. So the ODE equation change to be the explicit or implicit Euler equation. The implicit Euler equation actually is a nonlinear equations and can be solved by Newton iteration. Compare with Runge-Kutta method, the Newton iterative can constraint the error in each calculation step which can control the accumulation of error more precise. This character often ignored in the common exploration, and the advantage of Newton iteration is presented in some geometry mechanics references such as follows.

  1. Lee, M. Leok. Lie group variational integrators for the full body problem in orbital mechanics. Celestial Mech Dyn Astr (2007) 98:121–144.
  2. Lee, M. Leok. Time Optimal Attitude Control for a Rigid Body. 2008 American Control Conference Westin Seattle Hotel, Seattle, Washington, USAJune 11-13, 2008.
  3. Dmitry V. Zenkov, Melvin Leok, and Anthony M. Bloch . Hamel’s Formalism and Variational Integrators on a Sphere[C].51st IEEE Conference on Decision and Control,Dec. 10-13, 2012. Maui, Hawaii, USA

Comment 3

You wrote

  1. The dynamics solution of planer pendulum

Please change number of this section previous was numbered as 3.

 Answer: The number is changed in the updated version.

Comment 4

In line

Figure 3 Planar pendulum

Not cited and explained in paper. Also you present this figure without explanation anything about this section.Please improve it

Answer: the figure is cited in the updated version as follows

In this chapter, a planner pendulum is used to explain the above theory. The planer pendulum is as in Fig 3 which consider its geometry.

Comment 5

Also next section was numbered as 3?

  1. Dynamics modeling of 3D pendulum

Please renumber it on 5

  1. Dynamics modeling of 3D pendulum

Please also describe of Figure 4 before it presentation in article

Comment 6

You wrote section 6 as

  1. Simulation

Please change number on 6

  1. Simulation

In this section you wrote

Supposing that the mass of the pendulum is 2kg, the rotation inertia is 0.48kgm2, the length of the planer pendulum is 0.6m, the initial angle is 0 rad, the initial angular velocity is 1rad/s, the simulation time is 50s. according to equation (43) and (46), the simulation results are as fig 5 to fig 8.

I propose use in paper  Fig. 5 to Fig. 8 and also please improve all “fig ” on “Fig. ”

Answer: the above problems are updated in the new version.

Comment 7

Fig 7 The comparison of the later 5 seconds 400

Fig 8 Comparison of two difference calculation

Figure 7 The comparison of the later 5 seconds 400

Figure 8 Comparison of two difference calculation

And next

Fig 9 The difference calculation results under reduced attitude

Fig 10 The results under reduced attitude and triangle function(1)

Please change on

Figure 9 The difference calculation results under reduced attitude

Figure 10 The results under reduced attitude and triangle function(1)

Answer: the above problems are updated in the new version.

Comment  8

I propose to use description of Figures 5 to 27 in column to clearly presents of relevant description to presented figures

Please also correct of bold letters used for

Fig 26 The angular velocity results under Lie algebra expression

Fig 27 The attitude results under Lie algebra expression

Answer: the above problems are updated in the new version.

Comment  9

You wrote conclusions

  1. Conclusions

According to the above exploration, the long time simulation character is influenced by the modeling method, the numerical method synthetically. Different modeling and numerical method has different sensitive to time step size. The simulation shows that under the nonlinear equation solving method, the dynamics model which conserves the Lie group structure has a good long time simulation character and low sensitive to the time step.

In my opinion is possible to prepare more relevant conlusions to presented article.

Answer: the conclusion is updated in the new version as follows . the renewed conclusion summarize the discovery of this paper to be 5 conclusions and make the contribution more clearly.

From the simulation results of planer pendulum and 3D pendulum, the conclusions can be summarized as follows.

In the long time simulation, the influence of error accumulation to the dynamics not only reflected on the amplitude, but also on the periodic law of the system.

The reduced attitude model of planer pendulum can conserve the long time simulation character under special time step. If the time step is not suit, the error accumulation may lead to the hysteresis of simulation. So the error accumulation speed is obviously influenced by the time step.

The complete Lie group dynamics model has similar simulation characters under different time step, which means that the error accumulation is not sensitive to the time step. The simulation results under Lie algebra don’t have this advantage.

In the simulation of 3D pendulum, the reduced attitudes and complete Lie group also has a low sensitivity to the time step in the dynamics calculation. the simulation results under the time step of 0.1s and 0.01s have not obvious distinguish.

 The dynamics result of 3D pendulum is lost its conservation under the Lie algebra which is determinate by Cardan rotation, which means that the custom expression of rotation in space don’t have the long time simulation characters.

According to the above conclusions, the long time simulation character is influenced by the expression of model and the time step simultaneously. The complete Lie group expression has the best long time simulation character and has the lowest sensitive to the time step both in the planer and 3D pendulum simulation because it conserves the complete geometry characters of the system.

Round 2

Reviewer 1 Report

Dear Authors,

I appreciate that your updates improve the quality of the article, which now is publicable.

Author Response

The English grammar problems in this paper is revised, the revised parts are marked by the blue color in the paper, thank you for your suggestion. 

Reviewer 2 Report

Article was significantle improved. I propose to accept for publication. Regards

Author Response

Some writting problems in this paper are revised and marked by the blue color. 
